# Influence of Immune Parameters after *Enterococcus faecium* AL41 Administration and *Salmonella* Infection in Chickens

**DOI:** 10.3390/life12020201

**Published:** 2022-01-28

**Authors:** Viera Revajová, Terézia Benková, Viera Karaffová, Martin Levkut, Emília Selecká, Emília Dvorožňáková, Zuzana Ševčíková, Róbert Herich, Mikuláš Levkut

**Affiliations:** 1Department of Morphological Disciplines, University of Veterinary Medicine and Pharmacy, 041 81 Košice, Slovakia; viera.revajova@uvlf.sk (V.R.); terezia.benkova@student.uvlf.sk (T.B.); martin.levkut@uvlf.sk (M.L.); emilia.selecka@student.uvlf.sk (E.S.); zuzana.sevcikova@uvlf.sk (Z.Š.); robert.herich@uvlf.sk (R.H.); mikulas.levkut@uvlf.sk (M.L.); 2Institute of Parasitology, Slovak Academy of Sciences, 040 01 Košice, Slovakia; dvoroz@saske.sk; 3Institute of Neuroimmunology, Slovak Academy of Science, 845 10 Bratislava, Slovakia

**Keywords:** beneficial microorganisms, *Enterococcus* spp., chicken, *Salmonella* spp., immunity

## Abstract

Immune response of day-old chicks infected with *Salmonella* Enteritidis PT4 and preventive administration of *Enterococcus faecium* AL41 were studied using hematology and flow cytometry of immunocompetent cells in blood, cecum, bursa and spleen for 11 days, and included 220 animals divided into four groups (n = 55). *E. faecium* AL41 was administered for 7 days to EF and EFSE groups and on day 4 SE and EFSE groups were infected with *Salmonella* Enteritidis. Values of monocytes at 4 dpi significantly increased in EFSE and lymphocytes at 7 dpi in EF groups. Blood CD3, CD4, CD8 and IgM lymphocytes improved in EF and EFSE groups and IgA in EF group at 4 dpi. Phagocytic activity of probiotic groups was improved in both samples. Cecal IEL and LPL lymphocytes showed at 7 dpi stimulation of CD3, CD4 and CD8 subpopulations in probiotic groups, especially in EFSE group, IgA IEL and IgA with IgM LPL in EF groups. Bursa Fabricii at 7 dpi presented overstimulation of IgG subpopulation in SE group, spleen CD3 and CD8 in EF and EFSE groups. *E. faecium* AL41 revealed the protective effect and positive influence on the local and systemic immune response in *Salmonella* Enteritidis PT4 infected chickens.

## 1. Introduction

Salmonella infection, especially in developing countries, represents a health and economic burden on society at a global level. The treatment of complicated forms of the disease is less and less effective due to the alarming rise of resistant Salmonella strains, the negative impact of antibiotics on the intestinal microbiome, diarrhea associated with antibiotic treatment, which is a stimulus for the growing need for alternative treatments, including probiotic microorganisms [1]. The importance of improving salmonella control in poultry farming and procedures for introducing probiotic strains is becoming increasingly necessary in order to eliminate the potential economic burden and infectious threat to humans [2]. The initial intervention in the transmission of the pathogen and in the control of salmonellosis infection could be through creating a stable microbiome of the digestive tract of poultry [3].

Probiotics, prebiotics and synbiotics can be used to modify the gut environment to prevent *Salmonella* colonization, invasion, multiplication and shedding. This is particularly important in young poultry, in which stable intestinal bacteria have not yet been established [4]. Adding beneficial bacteria to feed or water allows them to populate the intestine and competitively exclude (or at least reduce) pathogen colonization, or to act by producing antibacterial compounds, mainly bacteriocins, as well to negatively affect pathogen metabolism by increasing or decreasing enzyme activity and to stimulate immunity by increasing antibody levels and macrophage activity [5]. Probiotic microorganisms have the ability to balance proinflammatory cytokines while increasing the number of anti-inflammatory mediators, including IL-10 and TGF-β [6]. The administration of these feed additives has a positive effect on the level of immunoglobulins M and A. The percentage of total antioxidant capacity in serum has also increased [7].

Probiotic bacteria in the host gut play an important role in the development and maintenance of both mucosal and systemic immune responses. The local microbiome not only forms the host’s immunity, but also contributes to protection against pathogens and infectious diseases [8].

We studied salmonella infection using *Salmonella* Enteritidis PT4, non-typhoid serovar of species *Salmonella enterica*. *S. enterica* is Gram-negative facultative intracellular anaerobes that can invade a broad range of hosts, causing both acute and chronic infections by means of their ability to replicate and persist within non-phagocytic epithelial cells as well as phagocytic dendritic cells and macrophages of the host innate immune system. There are two broad categories of proinflammatory stimuli during Salmonella infection; pathogen-associated factors stimulating the innate immune system of the host and virulence-associated factors leading to host processes causing the disease pathology [9].

Non-typhoid infection caused by *S. enterica* serovars, in contrast to the typhoid type, is characterized by extensive multiplication in the intestinal lumen, induction of the inflammatory process in the cecum, limitation of transmission and replication in deeper tissues such as liver and spleen, especially in chickens older than one week. The pathogen is recognized by epithelial cells, resident lymphocytes, macrophages and heterophiles, in order to trigger an organized process to limit the spread of salmonella to deeper tissues. Activation of immunity leads to an influx of heterophiles, macrophages, B and T lymphocytes, causing changes in the total gene expression of the cecal lamina propria [10]. Depletion of heterophiles transforms *S*. Enteritidis from a gastrointestinal infection to a systemic infection [11]. Salmonella infection induces the generation of specific T-lymphocytes, CD4+ and CD8+ cells, which play an important role in protection during primary and secondary pathogen recognition [12]. The interaction between T and B cells is essential for the development of acquired immunity to salmonellosis, for example, during the vaccination process [13].

Our earlier study [14] with *S*. Enteritidis PT4 strain and *E. faecium* EF55 did not significantly influence the values of leukocytes and bloody or cecal lymphocyte subpopulations, but showed significantly decreased salmonella colonization in the liver, spleen, caecum and feces resulting from local intestinal relationships between the probiotic microbiome and salmonella.

A bacterial strain with probiotic properties *E. faecium* reduces the colonization and proliferation of enteric bacteria, including *S*. Enteritidis [15,16,17]. *E. feacium* can produce bacteriocins which, unlike antibiotics, have a relatively narrow killing spectrum and are toxic only to bacteria closely related to the producing strain [17,18]. In chickens, early preventive administration of *E. faecium* has been shown to reduce cecal colonization by pathogenic *S*. Enteritidis, promote small bowel development, its protective barrier [19,20] and stimulate innate and acquired immune responses [21,22].

Further research should focus on the potential mechanisms, efficacy and forms of probiotic administration in the treatment and prevention of salmonella infections. A closer look at the cellular and molecular levels of the pathogen’s interaction with the chicken after application of a particular probiotic strain may contribute to reducing the prevalence of salmonella in poultry.

The aim of this work is to bring some knowledge contributing to the understanding of the immune mechanisms of enterocin M producing probiotic strain *E. faecium* AL41, especially the possible prophylactic potential in salmonella infection.

## 2. Materials and Methods

### 2.1. Experimental Scheme

The 11-day experiment included 220 *Salmonella* free one-day-old broiler chickens of the Cobb 500 breed (Mach hydina Budmerice, Slovakia) divided into four groups (n = 55): C (control), EF (*E. faecium* AL41), SE (*S*. Enteritidis PT4) and EFSE (combined group). The chickens were placed in large wire cages of menagerie with a temperature of 32–33 °C (reduced by 2 °C in the second week) and a humidity of 58–60%, with free access to feed and water, in compliance with hygiene principles. The chickens were fed with a commercial diet (BR1, Čaňa, Košice, Slovakia). During 1 and 7 days of the experiment, the EF and EFSE groups were individually orally administered a probiotic strain *E. faecium* AL41 at a dose of 10^9^ CFU/0.2 mL of Ringer’s solution. Chickens from SE and EFSE were infected per os with a single dose of *S*. Enteritidis PT4 at a concentration of 10^8^ CFU/0.2 mL phosphate-buffered saline (PBS) on day 4 of the experiment. Moreover, control and SE group received the same amounts (0.2 mL) of PBS to ensure the same stress conditions during the application of the substances.

Sampling was performed on days 4 and 7 after salmonella challenge (dpi) from blood and 7 dpi from cecum, bursa and spleen. The chickens were humanely killed, after administration of intraperitoneal anesthetics, by decapitation and exsanguination. The experiment was approved by the Ethics Committee of the University of Veterinary Medicine and Pharmacy and the Committee for Animal Welfare of the Ministry of Agriculture of the Slovak Republic (Č.k.Ro-270710-221).

### 2.2. Bacterial Strains

Experimental infection was achieved using *Salmonella enterica* serovar Enteritidis phage type 4 (SE PT4, kindly provided by DVM Šišák F. from The Institute of Veterinary Medicine, Brno, Czech Republic) prepared by Lauková A., DVM, CSc. (Institute of Animal Physiology, SAS, Košice, Slovakia).

*E. faecium* EF AL4 producing enterocin M was prepared by Lauková A., DVM, CSc. (Institute of Animal Physiology, SAS, Košice, Slovakia) [23].

### 2.3. Hematology

To determine the total number of leukocytes 25 μL of heparinized blood (50 IU mL^–1^ in PBS, Zentiva, Czech Republic) was diluted with 475 μL of Fried–Lukáč’s solution (DMD, UVMP, Košice, Slovakia). Leukocyte counts were counted in 100 small squares of a Bürker chamber with a light microscope at 40× magnification (Nikon, Düsseldorf, Germany). In blood smears stained with Hemacolor (Merck, Germany) was evaluated 100 cells. The absolute number of white blood cells subtypes (G.L^−1^) was calculated according to the formula: total leukocyte count × proportion of leukocyte subpopulations (%)/100.

### 2.4. Flow Cytometry

#### 2.4.1. Imunophenotyping of Lymphocytes

Isolation of lymphocytes from peripheral blood, spleen and bursa was performed according to [24]. For isolation and purification of cecal intraepithelial (IEL) and lamina propria lymphocytes (LPL), we followed the method Solano-Aguilar et al. [25] in modification for chickens [26]. Bursal and spleen samples were taken in PBS (Sigma, Rödermark, Germany), homogenized and filtered through a nylon sieve (BD, Kelberg, Germany).

Direct immunofluorescence method with single and double staining was performed using primary labeled mouse anti-chicken monoclonal antibodies (Southern Biotechnology Associates, Inc., Birmingham, AL, USA). To the suspension of 50 μL lymphocytes (10^6^ mL^−1^ PBS) was added monoclonal antibodies in dilution by datasheet for CD3, CD4, CD8 (T-cells), IgA, IgM (B-cells). Double staining arrangement was followed: CD3PE 8200-09/IgM FITC 8310-02, CD4PE8210-09/CD8α FITC 8220-02, and single for IgA FITC 8330-02). For control samples, polyclonal goat anti-mouse FITC-conjugated immunoglobulin F(ab’) fragment (Dako, Denmark) in 1:50 dilution with PBS was used. After 15 min incubation in the dark at room temperature, 1 mL of PBS was added to the cells and centrifuged (1400 rpm for 5 min). The supernatant was aspirated and the sediment was diluted in 0.2 mL PBS included 0.1% paraformaldehyde, which allowed them to be stored in a refrigerator at 4 °C until measured by a flow cytometer.

Lymphocyte subpopulation analysis was assessed with a FACSscan flow cytometer (Becton Dickinson, Heidelberg, Germany) and a dot-plot histogram using the CellQuest Software version 3.3. We calculated the absolute values of individual lymphocytes according to the formula:absolute lymphocyte count × relative percentage of subpopulation/100.

#### 2.4.2. Phagocytosis Assay

The function of polymorphonuclear cells was assessed by flow cytometry using whole heparinized blood and commercial PHAGOTEST kit (Phagotest, Heidelber, Germany). The PHAGOTEST kit contained fluorescein (FITC)-labeled opsonized bacteria (*E. coli*-FITC) and necessary reagents to measure the overall percentage of granulocytes that ingest one or more bacteria per cell.

Cells were analyzed by flow cytometry using blue–green excitation light (488 nm argon-ion laser) and CellQuest Software version 3.3. Granulocytes were measured after gating the relevant leukocyte cluster using a scatter plot (lin FSC vs. lin SSC) after collecting 5000 leukocytes per sample and analyzing the green fluorescence histogram.

### 2.5. Statistical Analysis

Statistical analysis of data was performed using one-way ANOVA with Tukey post hoc analysis using the statistical program GraphPad PRISM version 6.00 (USA). Differences between the mean values for different treated groups were considered statistically significant at *p* < 0.05, *p* < 0.01, *p* < 0.001. Values in figures are given as means and standard deviations (±SD).

## 3. Results

### 3.1. Hematology

Hematological examination of the white blood count (Figure 1a–d) showed a tendency toward an increase in monocytes at 4 dpi in the EF and SE groups, in the EFSE group also with significance (*p* < 0.05) compared to control. Overstimulation of lymphocytes in both probiotic groups was recorded at 4 dpi; however, at 7 dpi the highest values showed the SE and EF groups compared to the EFSE group (*p* < 0.05).

### 3.2. Flow Cytometry

#### 3.2.1. Phagocytic Activity

Evaluation of granulocyte function led to a higher percentage of phagocytic activity (% FA) (Figure 2a) in the experimental groups EFSE, EF and SE compared to the control at 4 dpi (*p* < 0.01), as well as at 7 dpi with a dominance of EF groups, followed by the EFSE and SE groups. The FA index (IFA) was also increased (Figure 2b) with the highest values in the EFSE group, significantly at 4 dpi *(p* < 0.05), and insignificantly in EFSE and EF at 7 dpi.

#### 3.2.2. Peripheral Blood Lymphocyte Subpopulations

Immunophenotyping of peripheral blood lymphocytes (Figure 3a–e) showed at 4 dpi an insignificant increase in CD3+, CD4+, CD8+ and IgM+ subpopulations in EF and EFSE groups compared to control. IgA+ lymphocytes had the highest values in the EF group; however, at 7 dpi, the values of SE group overnumbered controls and probiotic groups in determining subpopulations except IgM+ cells.

#### 3.2.3. Cecal Intraepithelial and Lamina Propria Lymphocytes (IEL, LPL)

The results of cecal IEL (Figure 4a) showed an increase in the relative percentage at 7 dpi of CD3+, CD4+ IEL in the EF, EFSE groups and IgA+ in the EF group. The increase in CD3+, CD4+ and CD8+ cecal LPL was recorded at 7 dpi in the EFSE group and IgA+ with IgM + in the EF group (Figure 4b).

#### 3.2.4. Bursal and Splenic Lymphocyte Subpopulations

Examination of bursal lymphocytes (Figure 5a) showed at 7 dpi the highest values IgG+ cells in SE and EF groups with significance to EFSE (*p* < 0.05). IgM+ cells in SE and EFSE groups were significant to C (*p* < 0.01) and EF groups (*p* < 0.05). 

Splenic lymphocytes at 7 dpi (Figure 5b) revealed significant differences of CD3+ in EFSE and EF groups compared to the SE group (*p* < 0.05), overstimulation of CD4+ in EFSE as well as CD8+ in EF and EFSE groups. IgM+ cells of SE group outnumbered EFSE group (*p* < 0.05).

## 4. Discussion

The introduction of new probiotic strains into practice requires detailed research in terms of their genetic stability, properties, but especially a comprehensive analysis of the mechanism of their effect on the host. The present work focused on the mucosal and systemic immune response in chickens, which was monitored by measuring changes in immunocompetent cells induced after application of pathogenic strain *Salmonella enterica* serovar Enteritidis phage type 4 (SE PT4) in conjunction with probiotic culture Ent-M producing *E. faecium* AL41 strain in blood and selected lymphoid organs. Part of our experiment was also the evaluation of the phagocytic activity of granulocytes in the peripheral blood. Such probiotic additives can contribute to improving the growth and efficiency of feed in broilers, their total live weight, quantity and quality of laying hens and, last but not least, can eliminate the risk of transmitting the infection to the human host. These results can be thriving in further research into a potentially suitable probiotic strain as a more innovative form of prevention and treatment of infectious salmonellosis.

The study by Lauková et al. [23] reported a positive effect of the probiotic strain *E. faecium* AL41 during a 42-day experiment on a sample of ostriches in which *Salmonella* infection was suppressed. In addition, the study also points to sufficient colonization abilities of the strain in the digestive tract. Its suppression is based on the interaction between the colonizing probiotic strain and the pathogen. One of the ways to explain this competition is the inhibitory activity of the strain due to the production of the antimicrobial peptide—enterocin M. Our assumption is based on the existence of this bacteriocin, which Marekova et al. [24] isolated from *E. faecium* AL41 strain and pointed to its broad spectrum of inhibitory activity against other enterococci by creating a natural barrier against pathogens. The work of Ciganková et al. [26] also showed less damage to the intestinal epithelial mucosa after application of enterocin A of the probiotic strain *E. faecium* EK13. In addition, the inhibitory potential may have been influenced by the lactic acid produced by enterococci.

Although the immune system of chickens is immature during the first weeks, probiotic bacteria have shown potentiating of the innate immune system and stimulation of the adaptive immune response in several studies [26,27,28]. One way to stimulate the immune response is through toll-like receptors, which act as sensors of microbial infection and are critical for the initiation of inflammatory and adaptive cellular responses. The study by Karaffová et al. [29] showed that *E. faecium* strain AL41 in chicken intestine can positively regulate the expression of TLR receptors (TLR4 and TLR21) and activate the production of IFN-β and CD14 cells infected with *Campylobacter jejuni* CCM 6191. White blood cell counts in our experiment showed a tendency to increase the monitored parameters in experimental groups EF, SE and EFSE at 4 dpi compared to control, in monocytes with significance (^ab^
*p* < 0.05) in the EFSE group. When assessing the number of heterophiles, significance (^ab^
*p* < 0.05) at 4 dpi in the SE group compared to control C was observed. We also recorded significance in the absolute number of lymphocytes at 7 dpi with the highest values in the EF and SE groups compared to the combined group EFSE (^ab^
*p* < 0.05). An increase in heterophils in SE groups is important to prevent systemic infection. Depletion of heterophils changes *S*. Enteritidis from a gastrointestinal infection to a systemic infection illustrating their critical role in early immunity [30]. Monocytes improving in SE and SE groups probably led to the killing of salmonella. The interaction with macrophages and dendritic cells and salmonella is a key stage in the progression of systemic infection in particular [31]. For clearance of the bacteria, innate immunity, namely the complement system and phagocytosis by macrophages, neutrophils and dendritic cells, are the most critical responses against the bacterial pathogens, while IFN gamma and antibodies resulting from adaptive immunity also dramatically enhance the innate immune response [32]. Decrease values in combined groups may be explained by the administration of probiotics by modulating the immune system in the intestine, including decreased inflammation, increased antibody response and stimulated phagocytosis [33]. Furthermore, probiotics can reduce pathogen translocation across the intestinal mucosa by enhancing intestinal barrier integrity and inflammatory cell recruitment [34].

In the later stages of infection with extracellular bacteria, antigen-specific components of immunity, B lymphocytes and the production of IgM antibodies are stimulated. With the help of T lymphocytes and their products (mostly Th2 type) there is a subsequent isotype shift and production of affinity antibodies of the IgG1 or IgA class, which effectively opsonize the bacteria. Secretory IgA effectively protects the host against intestinal and respiratory pathogens and prevents their adhesion to the epithelium. Bacteria containing a polysaccharide capsule are able to directly stimulate B cells by aggregating their BCRs and inducing the production of IgM antibodies. These, when bound to bacteria, activate the classical complement pathway. In the body persist memory antibodies of the IgG class, specifically IgA, which has a protective role and memory T and B lymphocytes [35].

Immunophenotyping of peripheral blood lymphocytes showed no significant increase in CD3+, CD4+, CD8+ and IgM+ subpopulations in the EF and EFSE groups compared to control at 4 dpi. IgA+ lymphocytes had the highest values in the EF group. At 7 dpi, the highest number of CD3+, CD4+, CD8+ and IgA+ subpopulations was found in the SE group. IgM+ reached the highest values in the C and SE groups followed by the EF and EFSE groups. When examining the white blood cell counts and subpopulations of T-lymphocytes, we noticed a phenomenon in which in the combined experimental groups of EFSE the values of immunological components at 7 dpi versus 4 dpi were suppressed or kept at similar values. This is supported by other studies that have shown that early administration of the *E. faecium* strain may cause a decrease in cecal colonization of pathogenic *Salmonella* strains [23]. We assume that it is also caused by the production of antimicrobial specialized inhibitory agents, such as bacteriocins, whose activity correlates with improved intestinal barrier function against bacteria. It would be necessary to perform experiments on the presence of infection in the body of chickens in addition to monitoring immune parameters, when we could exclude or compare the presence of *Salmonella* with respect to days after infection and application of the probiotic strain. The infection did not appear to progress during 7 dpi as a systemic infection due to the beneficial effect of the probiotic strain in the process of competitive elimination, resulting in a 7 dpi decrease in monitored immune parameters in the blood.

The crop and cecum are the major sites of *Salmonella* colonization in chickens after oral administration [14]. The interaction between the intestinal immune system and the commensal microbiome in chickens begins immediately after hatching and leads to initial inflammatory processes, by increasing IL-8 expression. This results in the infiltration of heterophiles and lymphocytes into the lamina propria or intestinal epithelium and the homeostasis of the intestinal immune system. Infiltrated lymphocytes are further differentiated depending on the composition of the intestinal flora, either to decrease the ratio of αβ to γδ T cells in the lamina propria and intestinal epithelium, or to change αβ T-cell receptor repertoires [27]. In the comparison of intestinal LPL to IEL, γδ T-lymphocytes are represented in a smaller amount (10%) to αβ T-lymphocytes, and CD4+ subpopulation of LPL predominates to CD8+ IEL [36]. Our results confirmed this by flow cytometry of LPL determination at 7 dpi with the highest values of relative percentage in CD3+ and CD4+ lymphocytes. CD3+ lymphocytes showed clearly the highest values in the EFSE group followed by the EF group. In CD4+ lymphocytes, overstimulation was observed in the combined group, followed by the SE, EF and C groups. Further CD8+ LPL lymphocytes were the highest in the EFSE group. In contrast, IEL at 7 dpi shows a predominance of CD3+ and CD8+ lymphocytes with the highest values in the EFSE and SE groups. Cytotoxic lymphocytes (Tc-lymphocytes) are important in the immune defense against pathogens. Cecal CD4+ IELs were most pronounced in their increase in the EF and EFSE groups. CD4+ cells are helper T cells that can stimulate phagocytosis of macrophages and modify B cells antibody production [37].

Significance was recorded for IgA+ IEL in the probiotic EF group in comparison to the SE and the combined EFSE groups (^ab^
*p* < 0.05). IgM+ IEL showed a significant increase in the SE group compared to the controls (^ab^
*p* < 0.05). The levels of the IgA+ and IgM+ LPL subpopulations enriched the highest values in the EF group. The increase in IgA+ lymphocytes in the intestinal mucosa indicates the immunomodulatory ability of our probiotic strain, which could be increased at a later stage in the combined EFSE group, as is supported by several studies [38,39,40]. Beirão et al. [38] published the effect of *E. faecium* in newborn chickens infected with *Salmonella* Enteritidis and increased IgA production, which was even twice higher on day 20 after infection in the combined vaccinated group with the probiotic strain compared to the pure vaccinated group without *E. faecium*. Similarly, the other experiment [39] demonstrated the probiotic effect of *E. faecium* AL41 in chickens infected with *S*. Enteritidis PT4 in the gut by increasing the secretory IgA (sIgA) concentration, which was probably the result of the cumulative effect of both bacteria. The study of Bobíková et al. [27] demonstrated the beneficial effect of *E. faecium* AL41 on IgA mRNA expression by overstimulation on 7 dpi with SE PT4 by PCR measurement, as well IgA+ increase in both EF and EFSE groups on 4 and 7 dpi by immunohistochemistry in cecal mucosae. This finding depicts the use of more and more precise methods for the study of the probiotic strain effects.

*E. faecium* AL41 strain in cecum at 7 dpi presented immunomodulation of IEL T cells by stimulation of CD3+ and CD4+ cells in both probiotic groups (EF, EFSE), and B cells by overstimulation of IgA+ cells in the EF group. In LPL, all determined T cell subpopulations (CD3+, CD4+, CD8+) showed stimulation in the combined EFSE group, together with B cells presented by overstimulation of IgA+ and IgM+ cells in the EF group. The role in overstimulation of IgA by IgA-inducing cytokines TGF-β4 and IL-17 in *E. faecium* AL41 in *Salmonella* infection was published [41].

Phagocytic activity is an essential component of the cellular innate immune response and plays an important role in host defense against microbial infection. We studied the phagocytic activity by a test based on the measurement of the respiratory (oxidative) burst of granulocytes after their stimulation by inactivated *Escherichia coli* bacteria. Evaluation of granulocyte functions resulted in a higher percentage of phagocytic activity (%FA) in the experimental groups in the order of EFSE, EF and SE compared to the control at 4 (^ab^
*p* < 0.01) and 7 dpi, led by the EF group, followed by EFSE and SE group. The index of phagocytic activity was also increased with the highest values in the EFSE groups, significantly at the 4 dpi (^ab^
*p* < 0.05), but also in the EFSE and EF groups at the 7 dpi with a higher index compared to the C and SE groups.

The mechanisms of innate immunity are the first line of defense against pathogenic microorganisms. An integral part of this reaction is the production of nitric oxide synthase (iNOS), which has a strong bacteriostatic activity against intracellular bacteria [42]. The in vitro study on peripheral mononuclear blood cells at our department indicated upregulation of iNOS gene expression in the combined group EFAL41 + SE 48 h pi suggesting the bacteriostatic effect of *E. faecium* AL41 [43].

*S. enterica* serovar Enteritidis infection is characterized by bacterial translocation across the intestinal epithelial barrier, followed by transient splenic infection up to 1 week after infection. The numbers of colony-stimulating units of *S*. Enteritidis in the liver, spleen and intestine were reported to peak during the first week after inoculation and then to gradually decrease [44]. Splenic lymphocytes in our experiment showed significant differences in the CD3+ cells, especially in the probiotic groups EFSE and EF to SE group (^ab^
*p* < 0.05). CD4+ and CD8+ lymphocytes of probiotic groups showed only tendencies toward increase. In the IgM+ subpopulation the SE group outnumbered EFSE group (^ab^
*p* < 0.05). Splenic B cells from the marginal zone strongly bind polysaccharide antigens and rapidly differentiate into IgM-producing plasma cells [45]. Elevated levels of IgM+ lymphocytes in the spleen in the SE group indicate early *Salmonella* penetration into this organ and the development of a systemic immune response. This is important for preparing vaccines, and it is known that splenic IgM+ lymphocytes are detected 48 h after immunization [46].

The bursa of Fabricius is the primary lymphoid organ found only in birds and is important in the processes of humoral immunity and antibody production. Lymphoid stem cells migrate from the fetal liver to the bursa during ontogenesis. By differentiation, these stem cells acquire the properties of mature immunocompetent B cells [47]. In our experiment, the highest values of the IgG+ subpopulation in the SE group compared to EFSE (^ab^
*p* < 0.05) were reached at 7 dpi. We also recorded significant values in IgM+ lymphocytes in the SE and EFSE groups compared to C (^ac^
*p* < 0.01) and EF (^ab^
*p* < 0.05). In spite of a low number of IgA+ lymphocytes found by flow cytometry in all experimental groups, a trend was recorded toward an increase in both EF and EFSE probiotic groups by using the ELISA test [38]. This is in consensus with findings during vaccination of chickens showing improved anti-SE IgG and anti-SE IgA detected by ELISA in serum, intestinal and oviduct samples in vaccinated chickens when compared to unvaccinated [48]. Once the infection is under control, which happens approximately 2 weeks after infection, expression of IgY (avian equivalent of IgG) and IgA increases to facilitate *Salmonella* elimination from the gut lumen [10].

Our results showed that early preventive application of *E. faecium* AL41 has a beneficial effect on the mucosal and overall immune response of chickens. We confirmed an increase in phagocytic activity in probiotic groups and thus stimulation of innate immunity. Despite the significant immunomodulatory effect of *E. faecium* strain AL41, further in vivo studies are needed to monitor the progression and presence of *Salmonella* infection and the influence of other factors on the competitive action of pathogenic and probiotic strains.

## 5. Conclusions

*Enteroccocus faecium* AL41 has been shown to have a potential beneficial effect through its influence on mucosal and systemic immunity in broiler chickens. This is affected by the modulation of monocytes and lymphocytes, including both the T and B cell subpopulations, not only in the pure probiotic EF group but also in the combined group infected with *Salmonella* PT4. An increase in the phagocytic activity of granulocytes was also determined. We suppose the main role of CD4+ cells is that as effector T cells can stimulate by their cytokines innate immunity presented by phagocytosis, cellular immunity by CD8 improve and humoral immunity by modifying B cells antibody production. The obtained results are the basis for further study of this probiotic strain applicable in scientific research using a more innovative form of salmonellosis prevention.

## Figures and Tables

**Figure 1 life-12-00201-f001:**
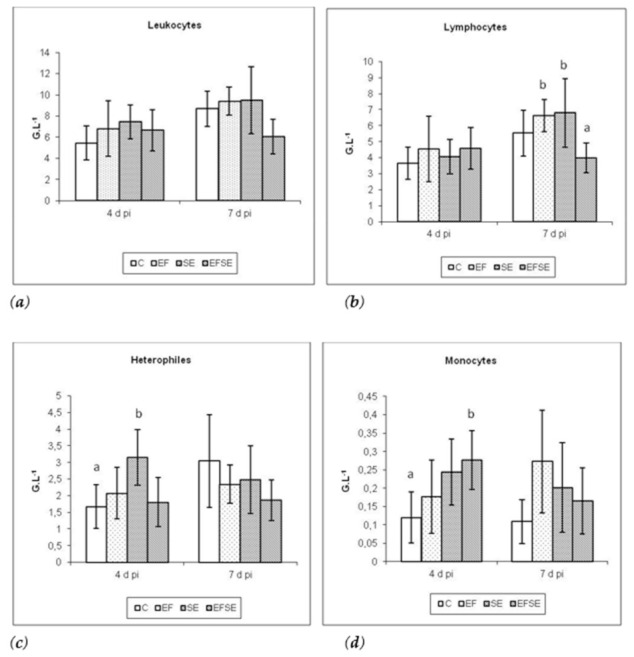
(**a**–**d**) Absolute number of white blood cell in G.L^−1^ (mean ± SD; ^ab^
*p* < 0.05) at 4 and 7 days post-infection (dpi). C (control), EF (*E. faecium* AL41), SE (*S*. Enteritidis PT4) and EFSE (combined group).

**Figure 2 life-12-00201-f002:**
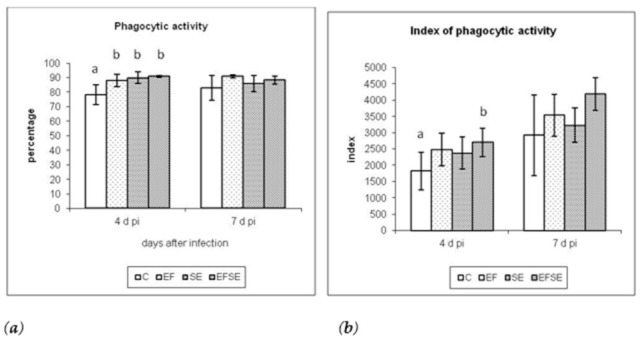
Granulocytic percentage of phagocytic activity (**a**) and index of phagocytic activity (**b**) 4 and 7 dpi (mean ± SD; ^ab^
*p* < 0.05). C (control), EF (*E. faecium* AL41), SE (*S*. Enteritidis PT4) and EFSE (combined group).

**Figure 3 life-12-00201-f003:**
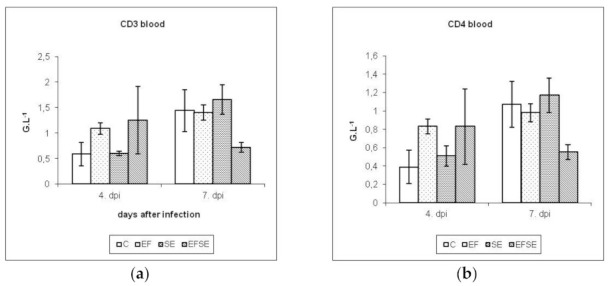
(**a**–**e**) Absolute number of lymphocyte subpopulations in the peripheral blood in G.L^−1^ (means ± SD). C (control), EF (*E. faecium* AL41), SE (*S*. Enteritidis PT4) and EFSE (combined group).

**Figure 4 life-12-00201-f004:**
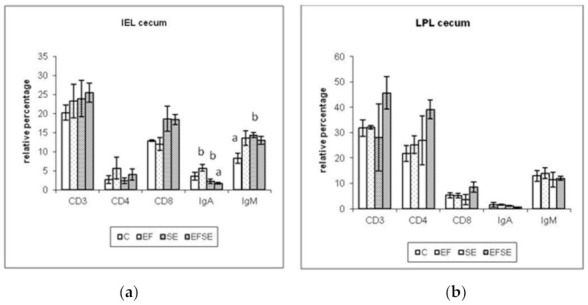
Relative percentage of cecal IEL (**a**) and LPL (**b**) at 7 dpi (mean ± SD, ^ab^
*p* < 0.05). C (control), EF (*E. faecium* AL41), SE (*S*. Enteritidis PT4) and EFSE (combined group).

**Figure 5 life-12-00201-f005:**
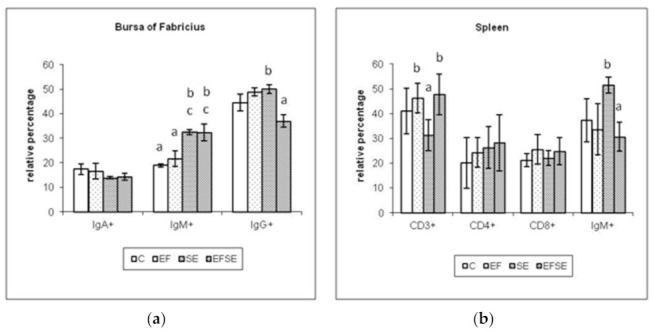
Relative percentage of bursal (**a**) and splenic (**b**) lymphocyte subpopulations at 7 dpi (mean ± SD, ^ab^
*p* < 0.05, ^ac^
*p* < 0.01). C (control), EF (*E. faecium* AL41), SE (*S*. Enteritidis PT4) and EFSE (combined group).

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
