# Peer review of "Influence of Immune Parameters after Enterococcus faecium AL41 Administration and Salmonella Infection in Chickens"

_life, 2022, doi:10.3390/life12020201_

Round 1
Reviewer 1 Report
There is some information that should be revised;

Author Response
Reviewer 1
Thank you for your comments. All changes are made in yellow color.
Basic reporting:
- “Enterococcus faecium” should be written in full at the first mention in manuscript and be written abbreviated as “ faecium” at subsequent mentions.
“Enterococcus faecium” was corrected in all text.
- Salmonella should be italicized, and “S” of Salmonella should be capitalized in whole manuscript.
Salmonella is now capitalized in whole manuscript.
- Line 57: non-thypiod -> non-typhoid
Line 57 “non-thypiod” was changed to “non-typhoid”
- Check the numbering order (Line 101: 4.1 should be 2.1 and other lines).
The numbering order was corrected.
Body:
- Authors should add a timeline of animal assay as a figure to make readers for easy understanding of your research. - I disagree, it is not necessary it was only two days of sampling
- Did control and SE group receive any control buffer during you administrated faecium to EF and EFSE group at day 0 and 7? Animals should be treated with the same stress condition during assay. – the sentence about it was added
- Did animals in the control group receive anything? Challenge with SE at day 4 or did not? What is the difference between control and SE group? – It is different that control received only PBS buffer and SE group received only one dose of S. Enteritidis, as mentioned in the text (part of Experimental scheme)
- Did author perform the natural occurring Salmonella count as a baseline? It may interfere your result, except your animals are germ-free. Why did authors not perform the baseline measurement of immune response? Is it sure that there was no infection before assay was conducted? Salmonella can be contaminated and infected in chickens from the broiler hatchery.
- Yes, we did, but results of it was published in our previous article (Žitňan et al., 2019, Muscle characteristics in chicks challenged with Salmonella Enteritidis and the effect of preventive application of the probiotic Enterococcus faecium, Poultry Science), anyway I add this information (see Experimental scheme)
- Why authors use 108 CFU/0.2 mL of SE in animal study? What is the common concentration of Salmonella in chicken growing on the farm? Is your selected concentration concordant to the real situation? Because we still use this dose, and many authors also use this concentration because it is sufficient to colonize the gut and elicit an immune response in chickens. However, this does not necessarily mean that it is the real concentration of salmonella found in chicken farms. We did not perform our experiment directly in chicken farming, where other environmental influences also apply.
- Authors should provide full information of bacteria strain preparation. I disagree because the full information about bacteria strain preparation is given as a reference (Lauková et al., 2015), thus avoiding repeating the same text
- For sample collection and examination, samples were pooled and investigated by replicates, or individual investigated (n=55/group)? Information should be given sufficiently and clearly. It might be related to the statistical analysis. Individual samples were investigated as is mentioned in the text (total 220 chicks were divided into 4 groups (n=55/ group)
- Did authors confirm the survivability of faecium during contact with gut conditions (low pH, bile salt, and/or enzymatic hydrolysis)? We did not directly confirm the survivability of E. faecium during contact with gut conditions but logically based on our results, we assume that this has happened.
- Why did authors not perform and show the result of Salmonella counting from cloacal swab during assay and from organs after sacrifice? It is very useful to correlate the result along with the immune responses. In this study, we did not look at the number of salmonella in cloacal swabs
- How many concentrations of Salmonella can induce the immune response? Authors should give more information in manuscript and correlate the result. The authors differ in opinion, but it is about 105 CFU. I disagree, because the article is not about what concentration is needed to elicit an immune response, but about the influence of a particular probiotic strain on the immune system of chickens during salmonellosis.
- Authors provide information of immune response sufficiently and clearly.
- Authors only discussed and compared their results along with 2 or 3 strains of faecium from their previous works. Authors should add information of other immunomodulator probiotic strains compared with the current work for avoiding inappropriate and unnecessary self-citations – I disagree, because in this part of the discussion not only our own previous works are discussed, but also other sources: Lauková et al. 2015; Mareková et al. 2007, Ciganková et al. 2004 ...
- In this study, authors performed the experiment of animal assay by administrating faecium before challenge with SE. The design of this assay is preventive treatment (EF and EFSE group). If you apply E. faecium after Salmonella challenge as exclusion group, does it affect the immune response similarly to your work? Should you give more advantages of your design and compare this with your work in discussion. I don’t understand what did you mean, because if you read precisely our manuscript, in that is written that EFSE group received also E. faecium (from 1 to 7 day of experiment) as well as S. Enteritidis (once time), but for deeper understanding was added in the experimental scheme: "Chickens from SE and EFSE "
- Authors should provide the antagonistic mechanisms of faecium on Salmonella sufficiently. However, the actual mechanism was provided by the antagonistic activity of bacteriocin M. Could authors show the result related to this study in-vitro test as supplemental file? I don't think it's necessary, because if someone wants to find it, they can look at our previous works, this study is not focused on the in vitro study.
- Could you give a detail why probiotics with antimicrobial production ability are more preferred to select than probiotics with strong adhering ability to epithelial cells for bacterial infection therapies. Even though the infection was initiated with the adhesion of pathogens on epithelial receptors. The detail should be added to your discussion section.
- Sentence about it was added into discussion section
Reviewer 2 Report
The manuscript is interesting and add to the existing knowledge on probiotics and its application in chickens . The manuscript is written well. What is the source of E.faecium? How many hour culture used in the study?
Author Response
Reviewer 2
The manuscript is interesting and add to the existing knowledge on probiotics and its application in chickens. The manuscript is written well. What is the source of E. faecium? How many hour culture used in the study?
Thank you for your opinion. Enterococcus faecium AL41 is an isolate of rabbit faeces and it was cultivated in MRS broth (300 ml; Merck, Darmstadt, Germany, pH 7) for 18 h at 37°C.